# Agricultural Social Networks: An Agricultural Value Chain-Based Digitalization Framework for an Inclusive Digital Economy

Ronald Tombe [1,2,*] and Hanlie Smuts [3,*]

1   Future Africa Campus, University of Pretoria, South St, Koedoespoort 456-Jr, Pretoria 0186, South Africa
2   Computing Sciences Department, Kisii University, Kisii 408-40200, Kenya
3   Department of Informatics, University of Pretoria, Pretoria 0083, South Africa
*   Correspondence: ronaldtombe@kisiiuniversity.ac.ke (R.T.); hanlie.smuts@up.ac.za (H.S.);
    Tel.: +254-710701596 (R.T.)

**Abstract:** Sustainable agriculture is the backbone of food security systems and a driver of human well-being in global economic development (Sustainable Development Goal SDG 3). With the increase in world population and the effects of climate change due to the industrialization of economies, food security systems are under pressure to sustain communities. This situation calls for the implementation of innovative solutions to increase and sustain efficacy from farm to table. Agricultural social networks (ASNs) are central in agriculture value chain (AVC) management and sustainability and consist of a complex network inclusive of interdependent actors such as farmers, distributors, processors, and retailers. Hence, social network structures (SNSs) and practices are a means to contextualize user scenarios in agricultural value chain digitalization and digital solutions development. Therefore, this research aimed to unearth the roles of agricultural social networks in AVC digitalization, enabling an inclusive digital economy. We conducted automated literature content analysis followed by the application of case studies to develop a conceptual framework for the digitalization of the AVC toward an inclusive digital economy. Furthermore, we propose a transdisciplinary framework that guides the digitalization systematization of the AVC, while articulating resilience principles that aim to attain sustainability. The outcomes of this study offer software developers, agricultural stakeholders, and policymakers a platform to gain an understanding of technological infrastructure capabilities toward sustaining communities through digitalized AVCs.

**Keywords:** resilience principles; climate adaptation; agricultural big data; society 5.0; inclusivity; internet of things; artificial intelligence; sustainability

## 1. Introduction

Social networks (SNs) are the different groupings of communities with shared practices, interests, and goals [1]. SNs in the agricultural value chain are heterogeneous and complex networks of many entities, including farmers, distributors, processors, and retailers. During the COVID-19 pandemic lockdowns, various digital platforms sustained the different SNs, including social healthcare engagements and [2] online teaching and learning [3]. In agricultural value chains, the COVID-19 lockdown effects severely affected the rural agriculture sector, especially smallholders [4].

Sensors and Internet of Things (IoT) usage in farmlands, mapping, and tracking technologies are revolutionizing farming systems and the management of the agricultural food system as the information moves from farmers to extension offices, traders, and consumers in the agricultural value chains [5]. The speedy growth of agricultural social network (ASN) entities has resulted in "agricultural big data" [6] characterized by the "5Vs" [7], that is, volume, variety, velocity, veracity, and value. Digital agricultural applications utilize various technologies, including IoT [8], data mining and analytics [9], blockchain [10],

big data and artificial intelligence [11], and sensor technologies [12]. Digital agricultural applications may integrate multiple big data sources such as remote sensing [13], farmers' inputs [14], machine and sensor data, public data, and other private-entity-held data [15]. Coble, Mishra [7] define agricultural big data as large, complex, diverse, longitudinal datasets generated from sensors, internet activities, emails, academic publications, videos, satellites, and other digital sources. For this study, we consider the term "agricultural big data" as a combination of technologies for advancing analytics in creating new ways of processing data timeously within agricultural social networks. The digital solutions relating to access to farming information, understanding consumer needs, and obtaining market information reside in the design and deployment of capable big data processing models. Sonka [6] groups agriculture big data analysis into the following clusters: collaborator search, farm management, recommendation, and expert systems.

Systems integrating cyberspace and physical space in precision farming aim to increase the efficiency of agricultural operational practices, especially for commercial products based on extensive knowledge management [11]. The cyber–physical systems are useful in monitoring and estimating crop yield via intelligent farm machines, sensor tools, and software that analyze the data collected from these devices. ASNs are important in determining the flow of information, resources, and services within the agricultural value chain. Agriculture is key to the world's economy, providing food, raw materials, and employment opportunities. However, the sector faces several challenges, such as the increasing demand for food, climate change, and the need for sustainable farming practices [16].

Innovations which leverage digital technologies such as IoT [17] and social capital [18] are beneficial in addressing some challenges facing the agricultural sector. Digitalization offers an opportunity to combine digital and social aspects, thereby providing a hybrid strategy for resolving and mitigating challenges in agricultural value chains [15]. Digitalization consists of technologies: the Internet of Things (IoT), big data, systems integration, sensors, robotics, ubiquitous connectivity, augmented reality, machine learning, blockchain, and artificial intelligence [11,15].

An inclusive digital economy relies on internet-based services and e-commerce applications [19]. Scholars report that the digital economy yields up to 3% of international job opportunities, which translates into 5% of the global gross domestic product (GDP) [20]. Hence, entrepreneurial innovation that fosters adaptive capacities of systems in uncertain and complex environments, such as an agricultural digital economy ecosystem, should be considered [21,22]. Digitalization impacts, disrupt, and causes transformations in the AVC [23,24]. The global challenges resulting from the economic effects of climate change [25], the increase in world populations, and advances in science and technology [26] present further reasons for transdisciplinary research, which require multistakeholder collaborations in resolving the greater challenges to planet Earth. Transformations in social changes further aggravate this situation due to the post-COVID-19 pandemic realities that have resulted in the shifting of data storage from traditional to digital forms [27]. In the context of sustainability in agriculture, efforts are being made in the digitalization of agri-cooperatives [28], indigenous agricultural practices in rural areas [29], smart agriculture [30], and supply chain management [31]. Resilience is key to attaining sustainable food systems [32]. Vroegindewey and Hodbod [33] define resilience as the capability of any system to continuously offer services and adapt to ongoing changes. Vroegindewey and Hodbod [33] propose a resilience framework for the agricultural value chain based on seven principles, i.e., fostering complexity, systems integration, broadening participation, managing connectivity, adaptive systems thinking, managing slow variables, encouraging learning, and enabling feedback.

However, the digital ecosystem still needs a clear implementation framework of resilience principles. Therefore, this research aims to address this deficit by proposing a digitalization framework for an inclusive digital economy ecosystem. The contributions are as follows:

1. *Types of ASNs*: This research unearthed the different types of social networks within the agricultural value chain, as well as the role of social networks in facilitating information sharing and knowledge transfer.
2. *Agricultural value chain digitalization*: This research established the potential of ASNs in the digitalization of the agricultural value chains to enhance the efficacy of the different stakeholders involved in the digital ecosystem.
3. *AVC digitalization conceptual framework:* This research developed a digitalization conceptual framework for the AVC digital economy ecosystem while utilizing the stakeholders' social networks' practices model.
4. *A transdisciplinary approach to AVC digitalization*: This research suggested a transdisciplinary approach to the digitalization of the AVC based on multistakeholder engagements in solving social and economic challenges.

The rest of this paper is structured as follows. In Section 2, we provide a comprehensive literature review. In Section 3, we present our research approach. In Sections 3.1 and 3.2, we present the content analysis and case study, respectively. Section 4 presents a conceptual digitalization framework for the agricultural value chain digital economy ecosystem. Section 5 discusses the research findings, and Section 6 concludes this paper.

## 2. Literature Review

### 2.1. Agricultural Social Networks

In the context of the agricultural sector systems, both in developing and developed countries, social networks are essential drivers of innovation and collaboration in climate change adaptation [34]. For example, Albizua, Bennett [35] establish that farmers in Spain innovate by following the practices of their successful contacts. Agricultural stakeholders, including farmers and government extension officials, make decisions in the context of an agroecological system to adapt to ecological, social, and economic changes. Stakeholders in agriculture adapt to these changes, and sustainability collaborations aim to build social networks (SNs) that facilitate innovation, learning, and partnerships. SNs often refer to 'social capital', implying that social processes such as partnerships and learning enable human societies to adjust to dynamic and complex circumstances [36]. Table 1 highlights typical applications of social networks in agricultural value chains.

**Table 1.** Applications of social networks in agricultural value chains (Authors' analysis).

| Social Networks Applications | Examples |
| --- | --- |
| Information sharing | Sharing of agricultural knowledge and information between farmers [37,38], researchers [39], and other stakeholders [39] occurs via social networks |
| Networking and collaboration | Stakeholders, including farmers, traders, extension officers, and researchers—from different locations—collaborate and share resources for innovations [40] |
| Market access | Farmers' access to larger markets [41,42] |
| Access to extension services | Extension services to farmers, including weather alerts, crop management advice, and marketing tips [43,44] |
| Crowd-sourcing data | Social networks are essential for collecting data on agricultural practices, weather patterns, and market trends [45,46] |
| Economic Empowerment | Digital innovations for women's economic empowerment [46] |

### 2.2. Agricultural Value Chain Sustainability Challenges and Mitigation Strategies

Agriculture value chain management has challenges due to factors such as the post-COVID-19 economic effects of climate change [25], inefficiencies in farming information, difficulties in accessing markets, and unreliable access to extension services [47]. Additionally, the supply chains of food systems are a daunting task owing to the need for better control and management systems to deal with varying supply and demand fluctuations, perishables, and the tight requirements for food safety standards and sustainability goals.

Digitalizing the agricultural value chain is critical to resolving these management challenges of food systems. The rapidly growing world population is putting pressure on food systems, and the effects of climate change further aggravate this situation [26]. Kaur [8] proposed a coordination strategy to address logistical barriers to enhance food security and mitigate climate change effects by adapting IoT-enabled agriculture value chain systems. Kaur [8] developed a Fuzzy-TISM IoT-driven sustainable food security system, which comprises different aspects, including e-farm marketing, traceability based on blockchain technology, quality control based on sensor technologies, smart contracts, crop insurance based on mobile app services, information sharing across networks, and e-governance in policy improvements. Other studies show that information communication technology innovations, particularly IoT, enables applications to have the potential to minimize food wastage and optimize distribution networks via a digital ecosystem, a concept referred to as digitalization [23,48]. Digitalization is a social–technical procedure of utilizing digital innovations in scaling operations, and improving efficiency and turnaround times in performing transactions [15]. Thus, digitalization can result in minimizing the carbon emissions of the entire food supply chain. AVC digitalization can yield innovative IoT-enabled digital systems to enhance productivity, food security and supply chain management, knowledge management, and the incomes of indigenous communities that collaborate through various agricultural social networks.

*2.3. Digital Technologies and Agricultural Value Chains in the Context of Society 5.0 Sustainability*

Digital technology provides avenues for services and new opportunities in achieving sustainability and transformation of the agricultural value chains [24]. Digital technologies include networks such as 4G and 5G, cloud computing, and the Internet of Things (IoT), which are rapidly becoming integrated into virtually all agricultural processes with a focus on enhancing efficiency, effectiveness, and sustainability [17]. IoT, in particular, provides farmers with digital solution tools that address numerous challenges in agriculture [11]. For instance, farmers connect to their farms at any time from anywhere through IoT-enabled devices [49]. The concept of Society 5.0 combines digital technologies with social innovation to promote a human-centric society [50]. Society 5.0's salient features, consisting of technology, systems, and people, as well as the interaction among these features, promote the digital economy in the agricultural sector. Table 2 summarizes the features of Society 5.0 that envision innovative human-centered smart applications that are knowledge-driven to meet sustainable solutions to differentiated physical and social needs in addressing societal challenges [51].

**Table 2.** Features of Society 5.0 for an inclusive digital economy in the agricultural value chain (Authors' analysis).

| Feature | Description |
|---|---|
| Human-centric approach | People are at the center of the development process to create sustainable, inclusive, and prosperous societies [52,53]. |
| Integration of digital technologies | Digital technologies, including artificial intelligence, the Internet of Things, and big data to solve social problems and enhance the quality of life [54,55]. |
| Collaboration | Stakeholders' collaborations, including government, industry, academia, and civil society, play a significant role in creating innovative solutions to social problems such as the lack of employment [56]. |
| Sustainability | Creating sustainable societies that balance economic growth with environmental and social considerations [57]. |
| Inclusivity | Innovation spurs the development of new business models to address social problems and create new opportunities [58]. |
| Digital empowerment | Empower individuals and communities by providing access to information and resources and promoting participation in decision-making processes [59]. |

Society 5.0 refers to a human-centered integration of the cyber and physical space, intending to balance economic advancement with the resolution of social problems [60]. Hence, to be inclusive and empower individuals and communities through the integration of digital technologies, stakeholders must collaborate and create digital economy solutions to achieve a sustainable agricultural value chain (SDG3) [61].

*2.4. Adoption of Information Communication Technologies in Agricultural Processes*

The availability of agricultural big data, coupled with the advances in digital technologies, is increasingly enabling the AVC [62]. With the integration of digital platforms, AVCs achieve increased responsiveness in accessing production extension services, distribution channels, and market demands more efficiently and cost effectively. Different stakeholders, such as customers, authorities, and agrifood businesses, play different yet integrated roles and, as such, contribute to the digitalization of the food system. Food systems' digitalization, in this context, refers to the application of technology to improve food harvesting, processing, distribution, and storage within the AVC [63].

The adoption of connected digital platforms in social and business fabrics is taking place quickly, ultimately changing how businesses operate globally. Agricultural big data with IoT and data mining analytic techniques provide significant insights for farmers, agricultural officials, and policymakers into crop yield production performance, fertilizer applications, soil mapping, weather data, and market trends [11].

Table 3 summarizes the technologies and phenomena enabling AVC digitalization. The primary purpose of technology application in AVC digitalization is to improve networking and associations along the AVC. In this context, the diffusion of technology is achieved through trial and error, resulting in a redesign of the solutions to ensure that the practical challenges of technology deployment at scale are reached.

**Table 3.** A summary of technologies enabling the digitalization of agricultural value chains.

| Enabling Digital Technologies | Technology Applications in AVC Digitalization |
| --- | --- |
| Internet of Things (IoTs) | Coordination, and logistics [64,65], quality management [66], smart farming [30] |
| Blockchain | Traceability of supply sources and transparency of food sources [10], food safety [67] |
| Artificial intelligence | Intelligent farm machines, greenhouse monitoring, drone-based crop imaging, social media and modernization of supply chains [11], precision agriculture [12] |
| Big data | Decision-making based on data and sustainable agriculture [68–70] |
| Augmented reality | Digital agriculture [71], precision farming [72] |
| System integration | Integrated agricultural farm management [15] |
| Machine learning | Digital agriculture and precision farming [73], crop disease detection, yield prediction, weed detection, water management, and crop recognition [74] |
| Edge computing | Big data processing [75,76] and smart AI applications in agriculture [77] |
| Cloud computing | Increased efficiency in the AVC [78] |
| Ubiquitous connectivity | Increasing connectivity along the AVC with the use of different digital devices and platforms to access and share agricultural information [79] |

## 3. Materials and Methods

The main objective of this research was to extract the roles of agricultural social networks in AVC digitalization to define an AVC digitalization framework toward an inclusive digital economy. We achieved this objective through an interpretive research paradigm and case study research strategy. A case study refers to "an empirical inquiry that investigates a contemporary phenomenon in depth and within its real-life context" (p. 240, [80]). This research applied two data collection methods. First, a systematic literature review (SLR), following the three-step process suggested by Rouhani, Mahrin [81], was undertaken to identify the corpus for automated content analysis. A Google Scholar

search was executed with the keywords "social network" and "agriculture value chain", creating a result set of 130 papers. We screened the corpus and excluded duplicate papers, papers written in languages other than English, and papers not relevant to our research. This resulted in a final set of 55 papers for detailed analysis to identify agriculture social network themes and concepts.

The overall objective of this paper was to develop an AVC digitalization framework for implementing resilience principles in enabling an inclusive digital ecosystem. A strategy to attain this goal was to understand the various ASNs within the AVC. The 11 themes identified via content analysis of the literature with the aid of *Leximancer* software fundamentally informed the design of the research questionnaires for the case study. Once the themes and concepts had been identified, we used a case study to collect data via a questionnaire, interviews, and focus groups [80]. The case study focused on indigenous vegetables in Kisii County, Kenya. Kisii County is an agricultural region in which food security systems are under threat due to the effects of climate change [82]. The data collection process we applied was similar to the approach taken by Abid, Ngaruiya [34] and Kiconco, Stevens [83]. The data collection methods utilized in the case study survey included questionnaires, focus group discussions, and interviews. Interviews were conducted with the agricultural extension officers to elicit information on (1) demographic information, (2) digital literacy levels, and (3) the application of digital devices in the provision of the extension of services.

Two sets of questionnaires were designed for two different categories of respondents, namely farmer and trader representative groups. For farmer respondents' representatives, the survey questions focused on eliciting information on the following aspects: (1) respondent demographic information, (2) digital knowledge sharing of farming information, (3) digital trade/marketing strategies, (4) innovation/value addition/processing, and (5) digital financial strategies. For the trader respondents' representatives, the questionnaire mainly captured the following aspects: (1) demographic data, (2) market information, including the pricing metrics, (3) the use of digital tools for business transactions, and (4) the digital financial credit facilities available for traders. For the focus group discussions, this research utilized the respective questionnaires of the farmer and trader categories to gather perspectives during different scheduled occasions. Research participants for the focus groups were identified through purposeful sampling, and simple random sampling was applied for the questionnaire and focus group discussions. The research participants included extension officers, farmers, and traders across all case study regions shown in Figure 1.

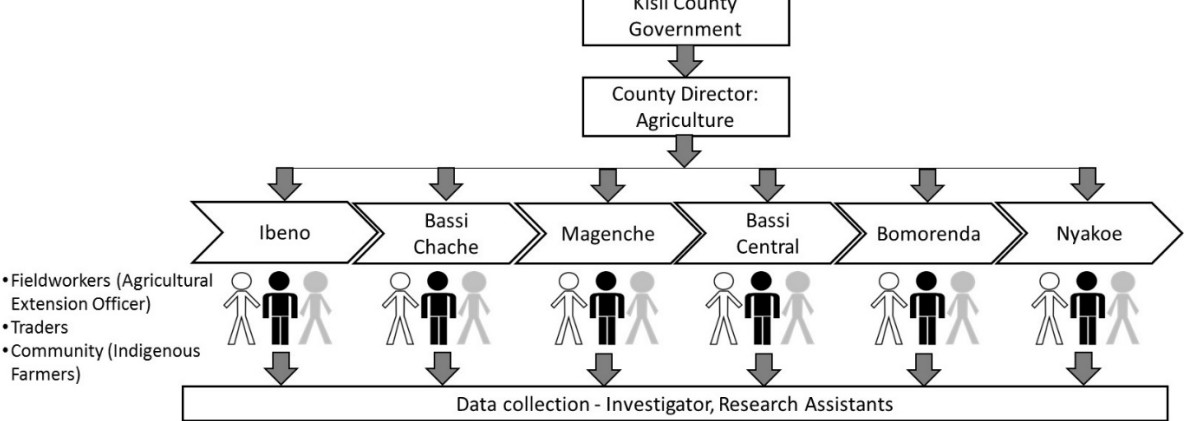

**Figure 1.** Case study regions indicating field movement plan and the actors involved in data collection.

The data collection was clustered into three classes, i.e., farmers, traders, and agricultural extension officers. Data were collected through the questionnaire and structured interviews in six regions/wards within Kisii County, namely Ibeno, Bassi Chache, Magenche, Bassi Central, Bomorenda, and Nyakoe, as depicted in Figure 1. Two focus groups

were held in the six regions. Table 4 provides a summary of the data collected and the participants' profiles.

**Table 4.** Summary of data collection methods, the types of stakeholder participants and the number of respondents per ward, and the totals per collection tool utilized.

| Data Collection | Stakeholders | Respondents per Ward | | | | | | |
| --- | --- | --- | --- | --- | --- | --- | --- | --- |
| | | Ibeno | Bassi-Chache | Magenche | Bassi-Central | Bomorenda | Nyakoe | Totals |
| Questionnaires | Farmer | 10 | 15 | 12 | 18 | 12 | 14 | **81** |
| | Trader | 15 | 18 | 16 | 10 | 12 | 11 | **82** |
| Interviews | Agr Ext. Officers | 1 | 1 | 1 | 1 | 1 | 1 | **6** |
| | Farmers-Group-Leaders | 2 | 2 | 2 | 2 | 2 | 2 | **12** |
| | Traders | 3 | 2 | 2 | 3 | 2 | 3 | **15** |
| Focus groups | Farmers | 1 | 1 | 1 | 1 | 1 | 1 | **6** |
| | Traders | 1 | 1 | 1 | 1 | 1 | 1 | **6** |

Different data analysis techniques were applied based on the data collected. The SLR corpus of 55 extracted papers was analyzed with automated content analysis software, and the case study data were collected through thematic analysis [84,85].

### 3.1. Automated Content Analysis in Identifying Agriculture Social Network Themes and Concepts

Content analysis (CA) examines the contents of a particular topic in the literature to identify feature patterns and themes [85]. Content analysis aims to show the information output with the circumstance in which they are generated [86] and varies between summative semantics and uncovering the content of illustrative communication [87]. Hence, content analysis can be both qualitative (targeted on understanding and interpreting) and quantitative (targeted on measuring and counting) [1,88].

Different software tools are applied in content analysis, particularly social network analysis [89,90]. Automated content analysis implements algorithms that use probabilistic models (such as terminological and concept mapping models) to reveal the dominant themes from the literature, their frequency, and their relationships [91]. The content analysis technique mechanism occurs in three phases: concept determination, concept definition, and text categorization [92,93]. In the initial phase, concept determination, individual words frequently occurring in the content are noted as concept seeds [92]. In the next phase, concept definition, the automatic content analysis software tool generates the class of words that form a concept using a concept mapping algorithm or topic model. This formation phase results in a set of dominant concepts, each formed by their vocabulary [94,95]. During the final phase, text categorization, the literature is categorized by the concepts located and defined based on the previous two phases. In text categorization, a text segment is analyzed for evidence of the occurrence of each concept [94]. After the execution of the three phases, the results are visualized through, e.g., concept maps, quadrant reports, social network maps, etc. These provide a synopsis of the literature analysis concepts and the relationships between the concepts [95]. Content analysis software tools show statistic and concept co-occurrence data, including summaries of concept and co-occurrence frequency between the concepts, offering quantitative illustrations of the prominence of particular concepts and the power of relationships between the concepts [96]. Additionally, the content data analysis tool reliably processes textual material, compared to humans, who may intuitively assign meanings to texts [97]. These demonstrate the reliability of automatic content data analysis tools [88].

In this research, we conducted the automated content analysis with *Leximancer* software, version 5.00.140 25 August 2021, articulating the methodologies of Watson, Smith [98], Kim and Kim [93,99]. *Leximancer* is a natural language processing software that uses

Bayesian theory. The *Leximancer* software tool applies unsupervised learning to determine the concept frequencies and their relationships. This process identifies the key concepts in text based on the interdependence of words, themes, and patterns from the data (www.leximancer.com (accessed on 23 January 2023)).

### 3.2. Case Study Data Analysis

The analysis of social networks in the AVC involved the identification of the different patterns and properties of the network. These patterns and properties included the degree of centrality of different actors in the network, the clustering coefficient, and the degree of connectivity between different nodes. Analyzing social networks in the AVC could help identify the different bottlenecks and inefficiencies in the chain.

The case study data analyzed were extracted from questionnaires, key informant interviews, and focused group discussions (FDGs) from six wards in Kisii County. The analysis process organized data into thematic areas through an inductive approach where patterns from the collected data guide its classification into themes [100,101]. In this case, our research developed two themes: (1) the stakeholders in the indigenous vegetable value chain and social network linkages within which information flows and services occur; and (2) the kind of information and services required by the AVC indigenous vegetable actors in facilitating transactions/services within their economic ecosystems and the extent of digital technology utilization in meeting their economic goals.

This research study further applied *Gephi* software version 0.10.1 202301172018 [102] for social network analysis to establish the linkages among stakeholders, that is, the data from individual stakeholders and county government representatives for every case study. Then, the analysis aggregated individual stakeholders into clusters. The goal was to establish clusters with similar patterns within the case studies.

### 3.3. Data Analysis Results

3.3.1. Results Based on Automated Content Analysis

The *Leximancer* software tool (www.leximancer.com) analysis for agriculture social networks identified 11 themes and 54 concepts depicted in Table 5.

**Table 5.** Themes and concepts of agricultural social networks (Source: *Leximancer* categorization).

| Themes | Hits | Concepts |
| :---: | :---: | :---: |
| Social | 5363 | interaction, networking, social media, spectrum of engagement, internet platforms, internet forums, social networks |
| Network | 4379 | link, group, social support, influence, centrality, structural characteristics, participatory programs, relationships, interactions, presence, actions |
| Agricultural | 4274 | research institutes, agribusiness, government, agencies, industry, extension systems, social networks, technology |
| Knowledge | 3979 | information, bridges, interventions, design, management |
| Farmers | 3494 | decision-makers, farmers, investors, traders |
| Systems | 2039 | services, collaboration, production, artificial intelligence |
| Groups | 1715 | clusters, households, farmers, networks |
| Data | 928 | big data, analytics, visualization |
| Model | 807 | infrastructure, machine learning |
| Food | 711 | supply chain, food ecosystem |
| Nodes | 489 | contacts, links, graphs, entities |

Figure 2 visualizes the relationships between themes and concepts.

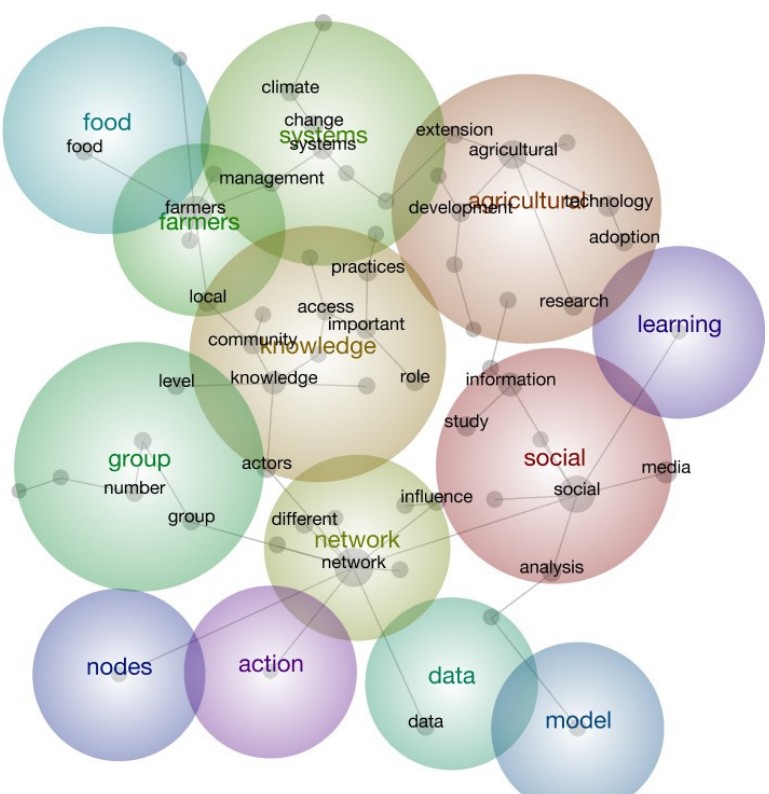

**Figure 2.** Themes and concepts maps of agricultural social networks (Source: *Leximancer* visualization).

The *Leximancer* visualization of ASN concepts in Figure 2 highlighted 11 themes. Table 5 expands further on the themes and also presents the concepts that are associated with each theme. Hits in Table 5 refer to the number of occurrences of a particular concept found within an associated theme. The *social* theme refers to the social aspect of networks, interactions, and the spectrum of engagement. *Network, nodes, and groups* highlight the structural characteristics of a group based on the relationships within the group and include clusters of farmers as part of the network. The *agricultural* and *farmers* themes confirm the context of the study as the research focused on the agricultural sector (agribusiness) in general and social networks within the agricultural sector specifically. The *knowledge* theme points to the management of information that may support the design of interventions based on lived experience. The *systems* and *data* themes incorporate services and the application of artificial intelligence (AI) in the context of big data and analytics, while *model* refers to the machine learning (ML) models that may be applied for analytics. The *food* theme includes the food ecosystem and incorporates the food supply chain associated with ASNs.

### 3.3.2. Models in Agricultural Social Networks

Based on the concepts and themes identified through the automated content analysis, Figure 3 presents the agricultural social network. The different colors (nodes in the form of circles) in Figure 3 show the various actors of agricultural social networks with types of information and services initiated via the linking edges. Note that the actor entity can either be a physical entity, for instance, a farmer, or a conceptual entity, such as a model.

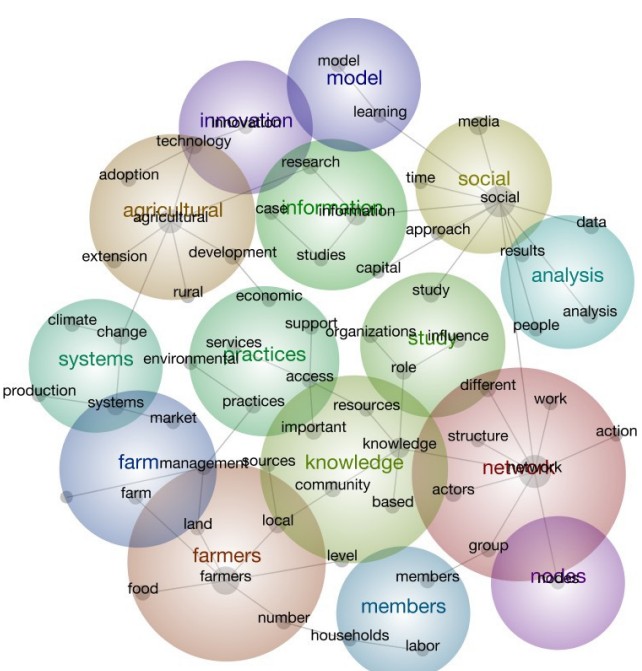

**Figure 3.** Agricultural social networks are complex and heterogeneous and integrate different entities (Source: *Leximancer* visualization).

Analyzing the ASN structures in Figure 4 shows the different clusters designated by color (e.g., red, blue, green, etc.), revealing the actors involved and the different types of information/service and knowledge exchanged between different actors in the chain. This is more pronounced in Figure 3: for instance, the farmers, practices, networks, and studies nodes intersect around the knowledge node. This knowledge may include market information, production practices, and processing techniques. Analyzing social networks in the agricultural value chain can help identify the different types of information and services that are most critical for the digitalization of the agricultural value chain processes and activities.

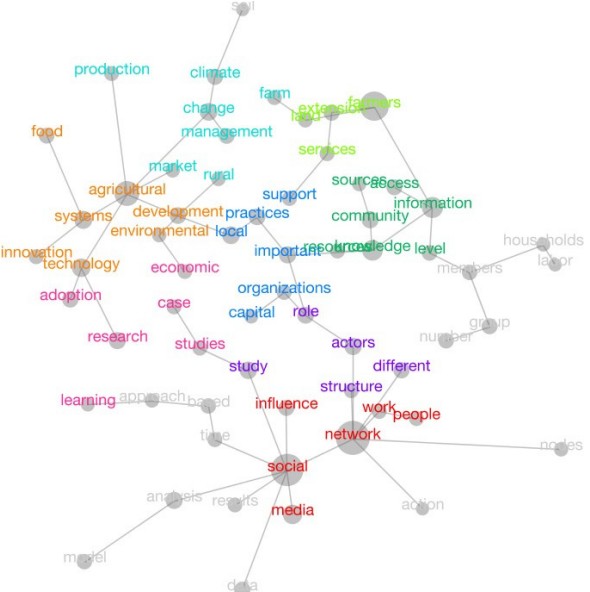

**Figure 4.** Agricultural social network structures (Source: *Leximancer* visualization).

### 3.3.3. Case Study Data Results

The data collected from the research survey revealed the main actors in the AVC, i.e., farmers/producers, traders, government entities (agric. ext. officers), the private sector, and financial providers. Based on these actors, we developed four global ASNs' digitalization strategies for their digitalization in attaining resilience and sustainability. The four global ASNs' digitalization strategies include digitizing farmer practices and experiences, digitizing trader services and products, digitizing agricultural extension officers' working ways, and attaining growth and collaborations through digital inclusivity.

*Digitizing farmer practices and experience*: This strategy involves using digital technologies to improve the farming practices of smallholder farmers, enabling them to increase productivity and efficiency. Using digital technologies such as mobile applications, remote sensing, and precision agriculture tools could help farmers optimize their use of resources, reduce waste, and enhance their decision-making capabilities. Farmers could access information on weather patterns, soil quality, crop varieties, and market prices through digital platforms. This information would help farmers make informed decisions and improve their yields.

*Digitizing trader services and products*: This strategy involves digitizing trading activities along the agricultural value chain, including marketing, distribution, and logistics. The use of digital technologies such as blockchain, IoT, and big data can help traders to track the movement of goods and services, enabling them to optimize their supply chains, reduce waste, and improve profitability. Digitizing trader services can also help to improve market access for smallholder farmers, enabling them to sell their products at better prices and reach a broader range of buyers.

*Digitizing agricultural extension officers' working ways*: This strategy involves digitizing *extension officers'* operational practices, enabling them to provide better services to smallholder farmers. The use of digital technologies, such as mobile applications and online training platforms, can help extension officers to provide timely and relevant information to farmers on topics such as crop management, pest control, and marketing.

*Attaining growth and collaborations through digital inclusivity*: This strategy entails the application of digital technologies to foster collaborations and partnerships along the agricultural value chain, enabling stakeholders to work together toward common goals. The use of digital platforms can help to connect smallholder farmers with traders, input suppliers, financial providers, and other stakeholders, enabling them to collaborate and share knowledge and resources. Through digital inclusivity, stakeholders can work together to develop innovative solutions to challenges such as climate change, food insecurity, and rural poverty.

The themes identified in the thematic analysis of the case study data were imported into *Gephi* to generate a network diagram (https://gephi.org (accessed on 12 February 2023)). These findings established clusters of farmers, farmer groups, traders/processors, the government, the private sector, research institutions, and financial service providers. Thus, we consider the various clusters as stakeholders in the agricultural value chain during data analysis of the ASNs. In Figure 5, stakeholders are depicted as nodes, whereas the relationships/linkages depict network maps in the case study research.

From the analysis of the findings, this research clusters the critical actors within the AVC into six groups, i.e., traders, farmers, government entities (agric. ext. officers), private sector actors (companies with initiatives that support small-holder farming initiatives under economic empowerment programs), and financial providers who advance agricultural credit facilities. This research utilizes the actors as the nodes, and the messages that flow between the actors are abstracted in the form of the edges between the involved actors (nodes) on the *Gephi* software to develop the network structure integrating the AVC actors.

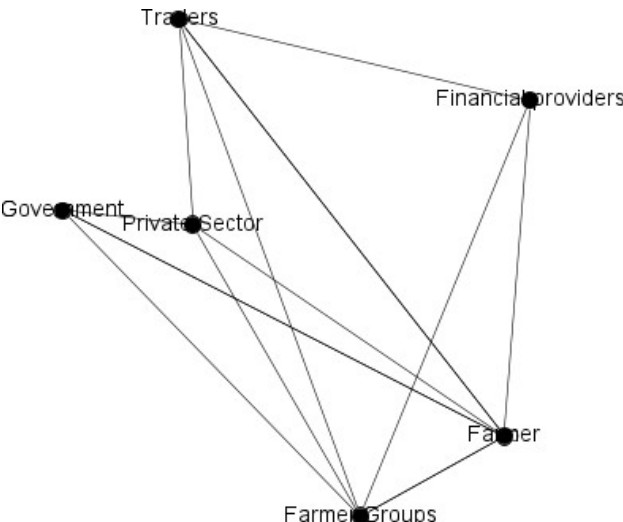

**Figure 5.** Stakeholders networks within the agricultural value chains for the Kisii County, Kenya case studies (Authors' visualization).

## 4. Digitalization of the Agricultural Value Chain in Enabling a Resilient and Inclusive Digital Economy

The overall objective of this paper was to develop an AVC digitalization framework for implementing resilience principles in enabling an inclusive digital ecosystem. Resilience is the capability of any system to continuously offer services and adapt to ongoing changes [33]. Two datasets were collected and analyzed, one from the literature and a second from case study data collected from farmers, traders, and agricultural extension officers in Kisii County, Kenya. We integrated the literature review and data from Sections 3.3.1–3.3.3 to consider the elements of an AVC enabling an inclusive digital ecosystem.

The food systems' value chains are crucial for livelihood sustainability, nutritional benefits, and other economic gains. Technological advances are revolutionizing agricultural sector operations, as discussed in Section 2.4. Based on the integrated data, Figure 6 presents a conceptual digitalization framework for the implementation of resilience principles in an AVC digital economy ecosystem.

Figure 6 shows that the AVC digitalization framework integrates different stakeholders' processes and practices via technology. These processes and practices include production systems, processors (value addition) distribution channels, and market segments across various agricultural value chains. Digital technology infrastructures and capabilities are the critical enablers of the digitalization framework for artificial intelligence solutions and applications that utilize agricultural big data for efficient decision-making. The critical components of the AVC digitalization framework include system integration for efficiency in information access; blockchain for enforcing transparency and traceability; ubiquitous connectivity in enabling social networks activities; and edge computing for big data processing, including data mining aided by machine learning business model for artificial intelligence applications for enabling access to financial services and ubiquitous connectivity market accessibility. All these technologies facilitate digital services empowering all stakeholders in this ecosystem. The subsections below provide more detail on the digitalization framework.

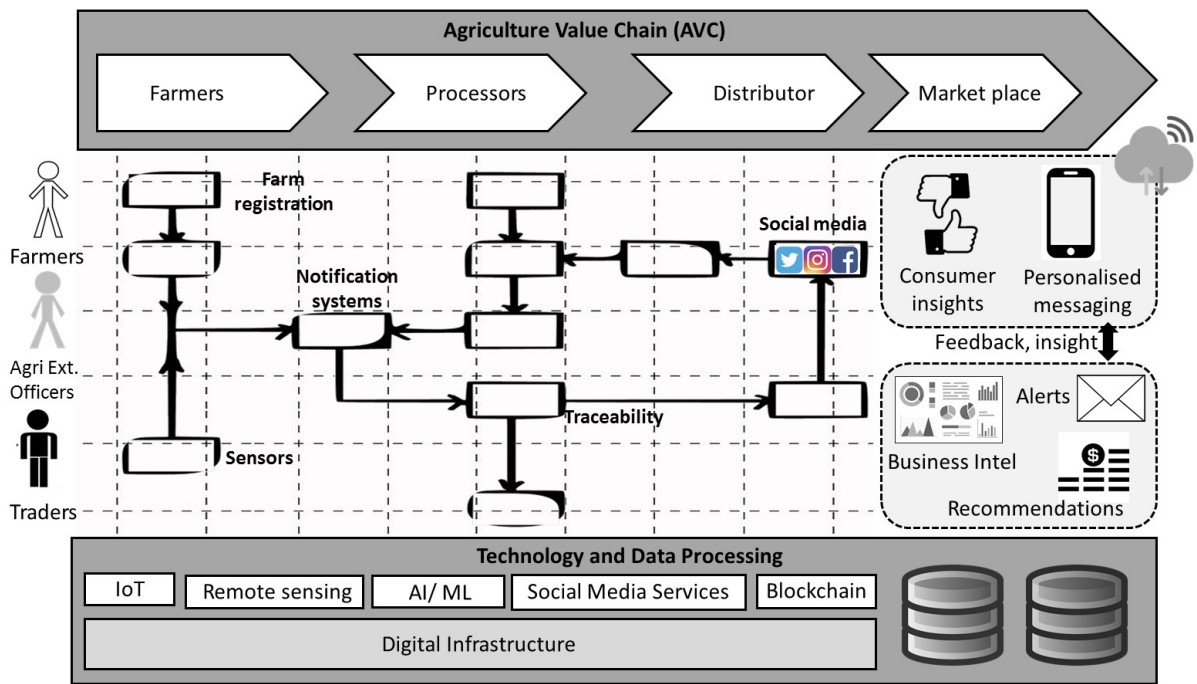

**Figure 6.** A conceptual digitalization framework for the agricultural value chain digital economy ecosystem (Source: Authors' visualization).

### 4.1. Systems Integration in Enabling Smart Farming and Efficient Decision-Making

Data processing facilitate seamless data, processes, and systems integration. These integrations can yield efficient services such as farm registration, implementation of the notification system, and social media services. All this is enabled by various technologies, including IoT, remote sensing, AI, digital technology, and machine learning. In smart farming, various components enable actors to collaborate in information-sharing practices and service provision. Agricultural data are heterogeneous as they have different formats, types, intentions, methods of data collection, and protocol devices. Innovative farming systems require timely and accurate actions and decisions. Therefore, there is a need for efficient platforms to enable farmers to take appropriate action promptly to enhance farming practices and, ultimately, increase productivity and profitability.

### 4.2. Blockchain for Enforcing Traceability in Supply Chain Management

Participants in the supply chain require assurance that products are safe and of high quality. Traceability is vital in supply chain management. Blockchain technology supports digitalizing the agriculture value chain by implementing transparent and secure digital systems. With blockchain enablement, AVGs could, therefore, verify and record legitimate transactions, highlight invalid transactions, coordinate transactions, and link the physical flow of goods to information flows.

### 4.3. Edge and Cloud Computing for Enabling Big Data and Governance

The digitalization of agricultural value chain practices will eventually give rise to big data ethical and privacy issues, prompting questions such as who owns the data and who is responsible for the big data governance requiring answers. Stable relationships among supply chain stakeholders, including farmers, enterprises, other logistic organizations, and retailers, enhance the agricultural supply chain's governance, strengthening the agriculture industry's competitiveness. The questions on data responsibility are very contextual. Therefore, the AI solutions will be situation-dependent, providing an understanding of data-driven processes and design systems that reckon with the challenges and outcomes of the decisions faced by humans.

### 4.4. Internet of Things in Facilitating Digital Financial Inclusion

Financial inclusion is fundamental in sustaining agriculture sector operations, economic growth, and sustainable employment. Digital financial inclusion is the provision of financial services digitally, especially to financially underserved groups of individuals at low cost [103]. Digital financial inclusion provides the means for capital formation, investment opportunities, and savings for individuals and enterprises. Technological innovations and digitalization greatly enhance digital financial services, inclusivity, and sustainable development. Consequently, digitalizing agricultural social network practices will mean inclusive economic growth among the communities involved.

### 4.5. Ubiquitous Connectivity for ASNs and Digital Markets

The digitization of economies is shifting the nature of businesses and changing how firms' digital markets compete. Using big data from agricultural social networks' online platform activities and processes provides ripe avenues for business analytics and intelligence in establishing new digital markets while broadening the market segments to launch agricultural products.

### 4.6. Interoperability and Scalability of Digital Systems

Interoperability has four broad categories: operational interoperability, system/technology interoperability, data/semantics interoperability, and legal interoperability [14]. The interoperability of digital systems/technology in operationalizing the exchange of standardized data formats within the established ethical and legal framework can enhance flexibility among stakeholders in agricultural social networks. Thus, they can collaborate in data and information sharing on farming practices, markets, market prices, and agricultural product distribution channels. Interoperability combined with IoT technology can scale the scope of digital systems applications from multifaceted perspectives, including increased financial access, inclusiveness of underserved agricultural populations, and access to broad digital markets regionally or globally.

## 5. Discussion

ASNs are essential for social–ecological adaptations and resilience in food systems. Digital information can enhance farm management by offering farm advisors and farmers timely and actionable intelligence, accelerating farm efficiency. In the digital era characterized by IoTs that seamlessly connect digital devices and vast amounts of agricultural big data results, it is important to envision how the various digital technologies work together to realize the vision of Society 5.0, provided in Section 2.3, which is creating inclusivity and sustainability in a digital ecosystem. Table 5 summarizes the fundamental themes, which are forms of knowledge obtained and generated from the analysis of ASNs for structural patterns that support agricultural value chain processes for sustainability. We can term the analysis of ASNs agricultural big data processing and from this, we establish the intersections between different themes as follows: (1) agricultural, innovation, information, and model; and (2) systems, farm, farmer, knowledge, practices, networks, members, and nodes, as depicted in Figure 3. The intersecting concepts of technology and systems suggest that social networks are enabled by technology and that this indicates the existence of cyberspace–physical systems for (a) innovation, (b) collaboration, and (c) boundary spanning. Our findings are consistent with the literature [36] that shows multiple pieces of evidence for all three patterns. Networks support the diffusion of innovation with short-path lengths and high levels of centralization, as depicted in Figure 4. Additionally, this research establishes that network structures supporting collaboration entail high levels of clustering. Thus, social networks facilitate collaboration, information sharing, and connectivity between specialized system components, leading to sustainability. The characteristics of ASNs discussed are critical in facilitating agricultural value chain services.

Finally, we utilized the findings from the case study research and the literature review analysis to develop and propose a transdisciplinary framework for the digitalization of the

AVC, depicted in Figure 7. The framework entails the following aspects: (1) community of engagements (farmers and traders), (2) government (agricultural extension officers), (3) the private sector (financial providers), (4) technology companies (digital technology providers), (5) research institutions (researchers and digital solution innovators), and (6) sustainability principles.

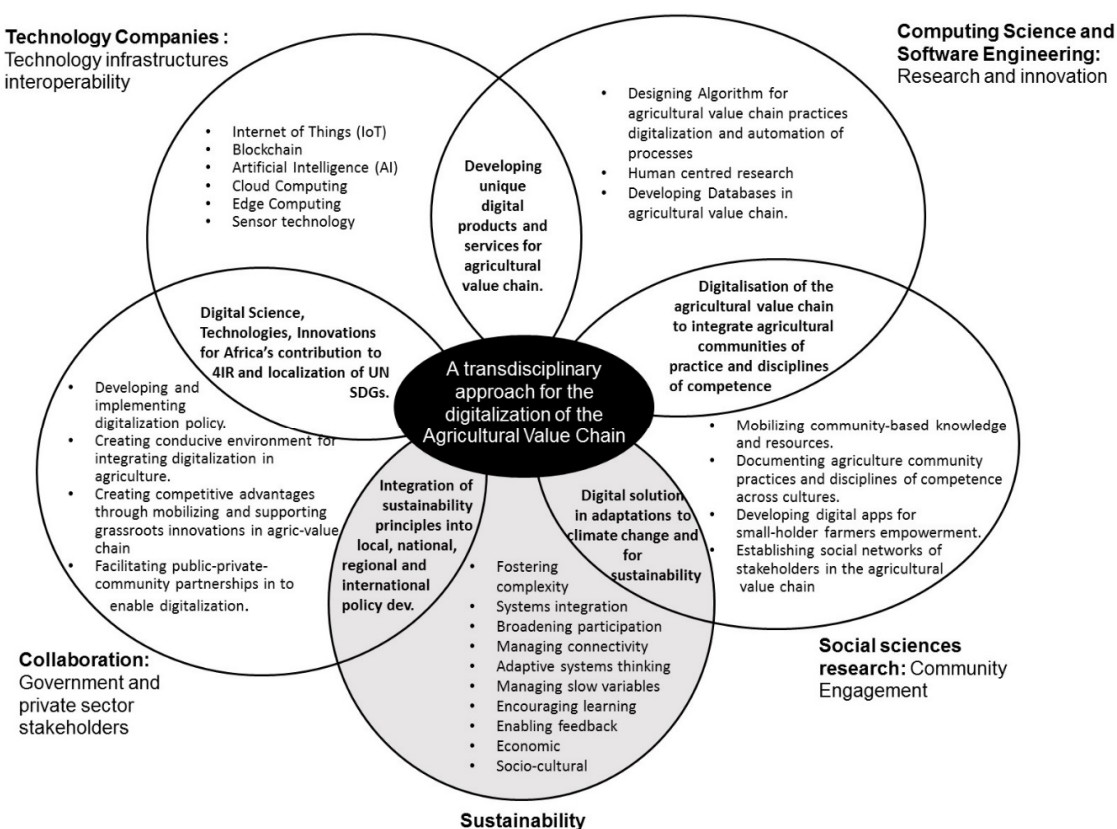

**Figure 7.** A transdisciplinary approach to the digitalization of the AVC (Source: Authors' visualization).

## 6. Conclusions

The ASNs' features provide powerful insights into the analysis and model of digital systems that integrate cyberspace and the physical space of the AVC ecosystem. Digital technologies such as IoT, big data, blockchain, edge computing, ubiquitous connectivity, and AI are the engines to drive current and future agricultural practices in achieving efficiency and timely decisions in a sustainable ecosystem.

This research utilized content analysis from the literature on ASNs using the *Leximancer* software tool. Content analysis revealed essential themes and concepts that gave rise to understanding different ASNs in the AVC context. This research also conducted case studies on the AVC of indigenous vegetables involving farmers, traders, and agricultural extension officers. The case study findings reveal links (information flows) between the agricultural value chain actors, which are the basis for services offered/received, thus facilitating the symbiotic nature of ASNs. This research clustered the different actors of the AVC and the information flow along the ASNs, which aided in the conceptualization digitalization framework of the AVC for an inclusive digital ecosystem.

Finally, this research developed a transdisciplinary approach to AVC digitalization that incorporates many ASN stakeholders (i.e., farmers, distributors, processors, and re-tailers). A proposed transdisciplinary research approach aims to provide a roadmap for implementing visions of Society 5.0 by creating a digital ecosystem that meets differentiated human needs in the AVC. The transdisciplinary research approach will be instrumental in implementing AVC resilience principles (fostering complexity, broadening participation,

managing connectivity, adaptive systems thinking, managing slow variables, encouraging learning, and enabling feedback). The two proposed frameworks will guide AVC innovations and develop digital solutions to enhance productivity, knowledge management, and the incomes of indigenous community farmers collaborating through different ASNs.

In our future work, we will utilize the digitalization framework to develop digital prototypes and digital solutions based on the requirements elicited from the case studies of the AVC.

**Author Contributions:** Conceptualization, R.T. and H.S.; methodology, R.T.; software, H.S.; validation, R.T. and H.S.; formal analysis, R.T.; investigation, R.T.; resources, R.T.; data curation, R.T.; writing—original draft preparation, R.T.; writing—review and editing, H.S.; visualization, R.T.; supervision, H.S.; project administration, R.T.; funding acquisition, R.T. All authors have read and agreed to the published version of the manuscript.

**Funding:** This research was funded by [Future Africa Research Leader Fellowship (FAR-LeaF) Programme at the University of Pretoria]. And The APC was funded by [Future Africa Research Leader Fellowship (FAR-LeaF) Programme at the University of Pretoria]. Grant number: G-20-57628.

**Institutional Review Board Statement:** Not applicable.

**Informed Consent Statement:** Informed consent was obtained from all subjects involved in the study.

**Data Availability Statement:** Data is unavailable due to privacy or ethical restrictions.

**Acknowledgments:** This research was made possible (in part) by a grant from the Carnegie Corporation of New York. The authors gratefully acknowledge support from the Future Africa Research Leader Fellowship (FAR-LeaF) Programme at the University of Pretoria.

**Conflicts of Interest:** The authors declare no conflict of interest.

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
