# Peer review of "Agricultural Social Networks: An Agricultural Value Chain-Based Digitalization Framework for an Inclusive Digital Economy"

_applsci, doi:10.3390/app13116382_

Round 1

Reviewer 1 Report (Previous Reviewer 2)

1.     The description of the abstract part should be concise, highlight the key points and contributions. It is suggested to add explanations on contributions.

2.     Some sentences in this paper are confused because of their lengthy structure. The author should write some simple sentences or sentences with simple grammar as much as possible, so that readers can understand the paper better.

3.     The punctuation of each sentence should not be ignored, that is, after each punctuation, a space should be left before a sentence is written. Please reexamine the relevant questions in the paper.

4.     The image title should be in the center position below the image. Please note the position of the title in Figure 1.

5.     The images in the manuscript should be beautified to make them more expressive.

6.     The position of the text in the picture needs to be adjusted to avoid obstruction and affect reading.

7.     The spacing and distribution of the tables in the manuscript should be adjusted to make them more harmonious. For example, Table 5 suggests slight modifications and attention to details.

The quality of language needs further improvement.

Author Response

Reviewer 1

Comment

Action

1.     The description of the abstract part should be concise, highlight the key points and contributions. It is suggested to add explanations on contributions.

Thank you for highlighting this. We have revised the abstract and made it more concise.

2.     Some sentences in this paper are confused because of their lengthy structure. The author should write some simple sentences or sentences with simple grammar as much as possible, so that readers can understand the paper better.

This comment is noted – the paper was language edited by a professional language editor and the certificate may be provided upon request.

3.     The punctuation of each sentence should not be ignored, that is, after each punctuation, a space should be left before a sentence is written. Please reexamine the relevant questions in the paper.

This comment is noted – the paper was language edited by a professional language editor and the certificate may be provided upon request.

4.     The image title should be in the center position below the image. Please note the position of the title in Figure 1.

We have updated and aligned the titles of tables and figures

5.     The images in the manuscript should be beautified to make them more expressive.

A graphical designer assisted with revising the images and ensuring that the look and feel of all images are consistent.

6.     The position of the text in the picture needs to be adjusted to avoid obstruction and affect reading.

A graphical designer assisted with revising the images and ensuring that the look and feel of all images are consistent.

7.     The spacing and distribution of the tables in the manuscript should be adjusted to make them more harmonious. For example, Table 5 suggests slight modifications and attention to details.

The language editor also ensured that table formatting was consistent.

Reviewer 2 Report (New Reviewer)

COMMENTS FOR THE AUTHOR:

This manuscript entitles Agricultural-Social-Networks: Analysis, Modelling and Conceptualization of AVC Digitalization Frameworks for an Inclusive Digital Economy is the product of a monumental effort to mention social networks-based study for importance of agriculture economy. The authors discuss the various aspect of ICT tools towards understanding the its advantages in agriculture economy.    Authors review the various articles in well-presented way and showing the importance of social networks in area of agriculture economy.  The case study and data analysis presented in well descriptive way and showing significant of each and every aspect of agriculture market entity. The paper is very use full for researches and student who are interested to work in area of agriculture domain.

Here mention the following suggested to improve the articles before going to publish.

1.      The articles mention more theorical aspect of information technology in area of agriculture. Suggested to authors reduce the more descriptive part.

2.      The case study presented by authors is not describe application of finding. Should include it in detail. 

3.      The data analysis approaches or techniques are not well described please improve as much possible.                

Author Response

Reviewer 2

Comment

Response

1.   The articles mention more theorical aspect of information technology in area of agriculture. Suggested to authors reduce the more descriptive part.

Thank you for the suggestion – we have reviewed and reduced the content where it was feasible.

2.      The case study presented by authors is not describe application of finding. Should include it in detail. 

Thank you for highlighting this – we have ensured that the data analysis and interpretation of the data is much more explicit.

3.      The data analysis approaches or techniques are not well described please improve as much possible.                

We enhanced and improved our description of the data analysis approaches.

Reviewer 3 Report (New Reviewer)

The manuscript by Tombe and Smuts proposed a digitalization framework for inclusive digital economy ecosystems. They claimed that no previous documented literature effort has taken into account the aspects of social networks in the digitalization of agricultural value chain activities and practices (Lines 97-99). I have to disagree, given that the information can be found in various publications, including Klerkx et al. (2019) https://doi.org/10.1016/j.njas.2019.100315 and Martens and Zscheischler (2022) https://doi.org/10.3390/su14073905.

I think the script contains numerous omissions and technical flaws, and even the title has some issues (e.g., too lengthy and AVC should be expanded).

The abstract, particularly the methodology and results part, is somehow vague, with a few typos (e.g., SGD3 should be SDG3 – line 8 and expanded at first mention). It is also too lengthy, with over 300 words. According to the journal guidelines, the maximum number of words for abstract is 200.

The research design is rather unclear in the Methodology section. Statistical analyses were insufficient - correlation and regression analyses should be done on the tested variables to understand their relationships/interactions.

The Discussion section is not precise and concise, with little support from the literature. The authors repeated what they mentioned in the Abstract and Introduction sections, without actually delving into the study's findings.

A few figures are unclear and difficult to read, particularly Figures 8 and 9. Figure 2 was adapted from Iaksch et al. 2021 - Reference No. 61 in the reference list but why Raheem et al referenced in the text - line 185?). Did the authors ask for the permission to use the figure?

I'd suggest the authors to have their script professionally proofread.

Author Response

Reviewer 3

Comment

Response

The manuscript by Tombe and Smuts proposed a digitalization framework for inclusive digital economy ecosystems. They claimed that no previous documented literature effort has taken into account the aspects of social networks in the digitalization of agricultural value chain activities and practices (Lines 97-99). I have to disagree, given that the information can be found in various publications, including Klerkx et al. (2019) https://doi.org/10.1016/j.njas.2019.100315 and Martens and Zscheischler (2022) https://doi.org/10.3390/su14073905.

Thank you for the references – our intention with this research was to add to this body of knowledge and we have re-phrased the point to reflect as such.

I think the script contains numerous omissions and technical flaws, and even the title has some issues (e.g., too lengthy and AVC should be expanded).

Many thanks - we have revised the title based on reviewer’s comments.

The abstract, particularly the methodology and results part, is somehow vague, with a few typos (e.g., SGD3 should be SDG3 – line 8 and expanded at first mention). It is also too lengthy, with over 300 words. According to the journal guidelines, the maximum number of words for abstract is 200.

We have shortened the abstract to align with the number of words and have addressed the SDG acronym and expansion.

The research design is rather unclear in the Methodology section. Statistical analyses were insufficient - correlation and regression analyses should be done on the tested variables to understand their relationships/interactions.

We have critically reviewed and improved our description of the method followed.

The Discussion section is not precise and concise, with little support from the literature. The authors repeated what they mentioned in the Abstract and Introduction sections, without actually delving into the study's findings.

Thank you for highlighting this – we have made it more explicit.

A few figures are unclear and difficult to read, particularly Figures 8 and 9. Figure 2 was adapted from Iaksch et al. 2021 - Reference No. 61 in the reference list but why Raheem et al referenced in the text - line 185?). Did the authors ask for the permission to use the figure?

A graphical designer assisted with revising the images and ensuring that the look and feel of all images are consistent.

We have requested approval to use the figure, however, we are still waiting for a response. We have therefore removed the image and have describe the intention of the visualization in the text.

Reviewer 4 Report (New Reviewer)

The paper is valuable and has considerable application values. It concerns a very important issue, the exploration of which requires an interdisciplinary approach.

Suggestions for Authors:

1.      Some excerpts from the "Materials and Methods" section should be moved to the results section.

2.      A section/subsection should not end with an object (table, figure).

3.      Objects (tables, figures) should not be in the immediate vicinity.

4.      Each object (table, figure) should be accompanied by at least a short commentary in the main text.

5.      In some cases, the way of the reference to the source should be corrected (e.g. lines 177, 286).

6.      I recommend reviewing the text and its appropriate correction (e.g. instead of "Growth Domestic Product" it should be "Gross Domestic Product" – line 73; instead of "x" a specific table number should be inserted – line 282).

7.      Bibliographic descriptions should be adapted to the editorial requirements of the journal.

8.      It would be advisable to move away from expressions personifying literature (e.g. lines 78, 125–126).

9.      I suggest converting the text to an impersonal form (e.g. lines 14, 299).

Author Response

Reviewer 4

Comments

Response

1.      Some excerpts from the "Materials and Methods" section should be moved to the results section.

Thank you for highlighting this – we have moved the results to the appropriate section.

2.      A section/subsection should not end with an object (table, figure).

Thank you for highlighting this – we have updated all such instances.

3.      Objects (tables, figures) should not be in the immediate vicinity.

Thank you for highlighting this – we have updated all such instances.

4.      Each object (table, figure) should be accompanied by at least a short commentary in the main text.

Thank you for highlighting this – we have updated all such instances.

5.      In some cases, the way of the reference to the source should be corrected (e.g. lines 177, 286).

We have updated and aligned this to the correct formatting.

6.      I recommend reviewing the text and its appropriate correction (e.g. instead of "Growth Domestic Product" it should be "Gross Domestic Product" – line 73; instead of "x" a specific table number should be inserted – line 282).

This comment is noted – the paper was language edited by a professional language editor and the certificate may be provided upon request.

7.      Bibliographic descriptions should be adapted to the editorial requirements of the journal.

We have improved the descriptions.

8.      It would be advisable to move away from expressions personifying literature (e.g. lines 78, 125–126).

Thank you for highlighting this – we have updated these references.

9.      I suggest converting the text to an impersonal form (e.g. lines 14, 299).

We have converted to the impersonal form.

This manuscript is a resubmission of an earlier submission. The following is a list of the peer review reports and author responses from that submission.

Round 1

Reviewer 1 Report

The article takes a very long time before the scientific question becomes clear.
Again and again, only general statements are made.
The scientific investigation is very thin. Here should be better and deeper research, especially since the statements are also different in different regions of the world.
The article needs to be comprehensively revised.

Author Response

Thank you for the insightful comments. We improved our manuscript accordingly. 

Reviewer 2 Report

This paper propose a digitalization framework, which is committed to improve the technical understanding of advanced agricultural technologies by all sectors of society. However, I have a few concerns:

1. The description of the proposed framework is not clear. And the model is lack of adequately described.

2. The analysis results are displayed graphically but lack of specific explanation. Eg. How to define the size of the circle representing different network models in Figure 3?

3. Please standardize the abbreviations of terms. After the abbreviation of a special term first appears, its full name should no longer appear in the following text.

4. The innovation and novelty of the article needs further improvement.

Author Response

Thank you for the insightful comments. we have improved our paper accordingly. 

Round 2

Reviewer 1 Report

Please note the comments from the first round.

Author Response

How we have improved the research paper generally based on the general reviewer comments

  1. Does the introduction provide sufficient background and include all relevant references?We have revised the introduction to give additional background information on our study while indicating the research gaps. At the same time, we provide appropriate citations and our research contributions. Please see the highlighted sections in yellow
  1. Are all the cited references relevant to the research?

We have reworked and improved the references accordingly to the best of our abilities.

  1. Is the research design appropriate?

We included a paragraph explaining our research design approach and provided citations in support of our approach. Please see the highlighted section in yellow.

  1. Are the results clearly presented?

The results generated by third-party software and within the short time required to improve this paper, we believe, are sufficient to advance our understanding of social network structures and to convey the research findings in the context of conceptualizing a digitalization framework.

  1. Are the conclusions supported by the results?

Revised the conclusion section and improved it accordingly. Please see the highlighted line in yellow.

Reviewer 1 comment

Comment one

 The article takes a very long time before the scientific question becomes clear.

Response

We have entirely revised the introduction to give additional background information on our study while indicating the research gaps while we provide relevant citations and the contributions of our research. Please see the highlighted sections in yellow.

Comment two

 Again and again, only general statements are made.

Response 

Thanks for the insightful comments. We revised our entire manuscript with the view of improving it, consequently, we minimized the use of general statements and we used articulating statements

Comment three

 The scientific investigation is very thin. Here should be better and deeper research, especially since the statements are also different in different regions of the world.

Response 

We appreciate this concern. We addressed this concern by providing a new section 5 with subsections 5.1 to 5.6 to incorporate more detail on the proposed digitalization framework. We also give more recent citations from different regions of the world. For instance, we include the following references:

  1. Qin, T., Wang, L., Zhou, Y., Guo, L., Jiang, G. and Zhang, L., 2022. Digital Technology-and- 409

Services-Driven Sustainable Transformation of Agriculture: Cases of China and the EU. Agri- 410

culture, 12(2), p.297. 411

  1. Nemchenko, A.V., Dugina, T.A., Shaldokhina, S.Y., Likholetov, E.A. and Likholetov, A.A., 2022. 412

The Digital Transformation as a Response to Modern Challenges and Threats to the Development 413

of Agriculture. In Smart Innovation in Agriculture (pp. 37-45). Singapore: Springer Nature 414

Singapore. 415

  1. Garske, B., Bau, A. and Ekardt, F., 2021. Digitalization and AI in European agriculture: a strategy 416

for achieving climate and biodiversity targets?. Sustainability, 13(9), p.4652.

  1. Abid, M., Ngaruiya, G., Scheffran, J. and Zulfiqar, F., 2017. The role of social networks in 427

agricultural adaptation to climate change: implications for sustainable agriculture in Pakistan. 428

Climate, 5(4), p.85.

  1. Mukhovi, S.M., Jacobi, J., Llanque, A., Rist, S., Delgado, F., Kiteme, B. and Ifejika Speranza, 554

C., 2020. Social self-organization and social-ecological resilience in food systems: lessons from 555

smallholder agriculture in Kenya and indigenous Guarani communities in Bolivia. Food studies, 556

10(1), pp.19-42.

Comment four 

 The article needs to be comprehensively revised.

Response

Thanks for this comment. We have revised the article comprehensively. Please see the highlighted sections in yellow.

Reviewer 2 Report

The manuscript is innovative for the proposed digitization framework, but overall there are still a few flaws as follows.

1. The fonts in Figure 1 and Figure 5 are not clear enough, especially in Figure 5.

2. The application format of the picture is confusing! "figure or Figure"?eg line 368 and 266. Please check the full text throughout.

3. line 283, "In figure 5 we present a". Pay attention to punctuation and grammar, and check the full text.

Author Response

How we have improved the research paper generally basing on the general reviewer comments

  1. Does the introduction provide sufficient background and include all relevant references?

We have entirely revised the introduction to give additional background information on our study while indicating the research gaps while we provide relevant citations and the contributions of our research. Please see the highlighted sections in yellow

  1. Are all the cited references relevant to the research?

We have reworked and improved the references accordingly to the best of our abilities.

  1. Is the research design appropriate?

We included a paragraph explaining our research design approach and provided citations in support of our approach. Please see the highlighted section in yellow.

  1. Are the results clearly presented?

The results generated by a third party software and within the short-time required to improve this paper, we believe that they are sufficient to advance our understanding of the of social networks structures and to convey the research findings in the context of conceptualizing a digitalization framework.

  1. Are the conclusions supported by the results?

Revised the conclusion section and improved it accordingly. Please see the highlighted line in yellow

Reviewer 2 comments

This paper proposes a digitalization framework, which is committed to improve the technical understanding of advanced agricultural technologies by all sectors of society. However, I have a few concerns:

Comment one

  1. The fonts in Figure 1 and Figure 5 are not clear enough, especially in Figure 5.

Response to the reviewer comment

Thank you for the insightful comment. We have addressed the concern by improving the image Dots per Inch (DPI) to 300. With some zooming in, the images should be clearer.

Comment two

  1. The application format of the picture is confusing! "figure or Figure"?eg line 368 and 266. Please check the full text throughout.

Response to the reviewer comment

Thank you for the insightful comment. We have formatted all the figure in the document use a consistent format manner, i.e. we use Figure.

Comment three

  1. line 283, "In figure 5 we present a". Pay attention to punctuation and grammar, and check the full text.

Response to review comment

Thank you for the insight. This issue has been addressed accordingly.

Round 3

Reviewer 1 Report

no additional comments